# Can Model Experiments Give Insight into the Response of the Soil Environment to Flooding? A Comparison of Microcosm and Natural Event

**DOI:** 10.3390/biology11030386

**Published:** 2022-03-01

**Authors:** Karolina Furtak, Jarosław Grządziel, Anna Gałązka

**Affiliations:** Department of Agricultural Microbiology, Institute of Soil Science and Plant Cultivation—State Research Institute, Czartoryskich 8, 24-100 Puławy, Poland; jgrzadziel@iung.pulawy.pl (J.G.); agalazka@iung.pulawy.pl (A.G.)

**Keywords:** EcoPlate, enzymatic activity, flood, hydrological event, microcosm experiment, model experiment, natural event, soil microorganisms

## Abstract

**Simple Summary:**

Model tests under laboratory conditions are very common in soil ecology and microbiology, but few of them are related to flooding, and comparison of the results of such an experiment with natural conditions is unprecedented. The present study investigated the basic parameters determining the biological activity of soil subjected to flooding under laboratory and natural conditions. The results obtained show that soil inundation under both natural and laboratory conditions significantly affects soil fertility and processes. The changes are more pronounced in the laboratory experiment than in the field conditions. Nevertheless, model studies are needed in environmental ecology and microbiology to predict changes under different stress factors, but their scale and pathway must be carefully planned.

**Abstract:**

Studies using soil microcosms are very common, but few involve flooded soils, and comparing the results from such an experiment with natural conditions is unheard of. In the present study, we investigated the biological activity of soil (pH value, dehydrogenases and phosphatase activities) and the metabolic potential (EcoPlate™ Biolog^®^) of soil microorganisms in three fluvisol subjected to flooding under laboratory and natural conditions. The results indicate that soil flooding under both natural and laboratory conditions affected soil pH, enzymatic activity and metabolic potential (AWCD, average well colour development) of soil microorganisms. Changes in these parameters are more pronounced in the microcosmic experiment than in the field conditions. Furthermore, depending on the characteristics of the soil (i.e., its type, structure, vegetation) some of the soil quality parameters may return to their preflood state. Microcosm studies are needed in environmental ecology and microbiology to predict changes due to various factors, but their scale and course must be carefully planned.

## 1. Introduction

The European Union Floods Directive defines a flood as a covering by water land which is not normally covered by water [1]. The European Academies’ Science Advisory Council (EASAC) conducts observations on climate change [2]. Researchers indicate that, although all types of phenomena—geophysical, meteorological, hydrological and climatic—show an increase in global occurrence, the highest increase was recorded for hydrological phenomena such as floods, mudslides and snow avalanches. In Europe, the flood risk has increased in the last years [3,4]. Over the past three decades, the number of extreme weather events, including hydrological events, in Europe has increased by 60% [2]. The forecasts show that climate change leads to an increase in the intensity of storms and flooding in Europe by 2100 [5]. It is also estimated to increase the sea level and the height of storm waves and thus to increase the frequency of coastal flooding events [5,6].

The water content of the soil affects the pH, the diffusion of solvents and gases and the availability of nutrients. Water also enables the migration of microorganisms in the soil and the diffusion of compounds between the cells of organisms and the environment and is part of hydrolysis processes, and its content determines the rate of mineralization [7]. Natural fluctuations in moisture associated with seasonal changes and precipitation are an important environmental factor in the metabolism of microorganisms. Recently, however, the frequency of floods and periodic flooding in Europe have been increasing, and drought periods have been prolonged [4,8]. Water stress caused by these phenomena impacts soil microorganisms [9].

As far as floodplains are concerned, few studies have been carried out so far. The soil under rice production is a well-known object of research in this area [10]. However, it is deliberately irrigated, and the homeostasis of such soils varies over many years; rice fields are not natural river floodplains. With the method based on 454-pyrosequencing in Bangkok floodplains (Thailand, flood in 2011), researchers have identified bacteria that reduce sulphates and nitrates [11]. In 2009, Unger et al. (2009) simulated the conditions of the flood in the greenhouse in order to determine the impact of the inundation on soil microorganisms [12]. A significant decrease of microbial biomass was observed during this experiment. However, the researchers themselves admitted that their study was too short to accurately estimate how much flooding affects the soil microbial community. Argiroff et al. (2017) have shown that there are relations between the composition of the microbiological community and the floodplain frequency, but the authors indicate that this subject has still not been sufficiently analysed [13]. After flooding in Poland in 2010, analyses were performed of the content of particular elements in soil from the municipality of Wilków (Lubelskie Voivodeship) including heavy metals, content of pesticides and PAHs (polycyclic aromatic hydrocarbon) [14].

The microcosm (small-sized, laboratory experiments) and mesocosm (medium- or large-sized, usually conducted outdoors in order to incorporate natural variation) are commonly used experimental models in biology sciences: ecology, microbiology, toxicology, etc. [3,8,15,16,17,18]. Microcosms have become a major research tool in soil ecology and have added considerably to our ecological knowledge [19]. They are used to simulate and predict the behaviour of natural ecosystems under controlled conditions [17]. They can be useful for analysing the influence of particular environmental elements or other factors on a given ecosystem. A pioneer of such research was Odum, E.P., who used bottles or fish tanks to create small closed or open ecosystems, which have been called microcosms [20]. The use of micro- and mesocosms experiments in biological, ecological research has shown light on many research problems hitherto impossible to address due to the high costs and lack of intervention in whole ecosystems [21]. However, they do not allow 100% reproduction of natural conditions. The soil in such an experiment has no contact with the bedrock; the influence of atmospheric conditions (e.g., wind, radiation, pressure), slope and other naturally occurring properties is eliminated. Moreover, the smaller the scale of the system, the less the data relate to reality (Figure 1). At the same time, such an experiment excludes other environmental factors (as well as human activity), and the observation of the influence of only a selected factor—e.g., high soil moisture—on the soil environment. This is important due to the complexity of the soil environment and its interactions. Many studies to date only rely on soil analyses after a hydrological event and neglect soil analyses before and during flooding [22,23,24]. The model experiment allows the soil to be observed throughout the stress period.

In this study, we set out to compare the results obtained from a microcosm experiment on simulated flooding with a natural hydrological event. This study was possible because one year after our laboratory experiment, exactly the same area from which we took samples for the experiment was flooded as a result of a river leaving its channel. The aim of our study was to analyse whether the biological activity of soils after 7 days of flooding and after the receding of water would maintain a similar trend in the model experiment and under natural conditions. Among the parameters analysed were enzymatic activity, pH, moisture content and metabolic potential of microorganisms using the EcoPlate™ method.

## 2. Materials and Methods

### 2.1. Soil Sampling Site

Three different soil classed as fluvisol [25], located in the Vistula River Gorge of Lesser Poland in Wojszyn Lubelskie Voivodeship (F1—51°20′03.4″ N 21°56′43.2″ E) and Janowiec Lubelskie Voivodeship (F2—51°19′29.9″ N 21°55′19.2″ E; F3—51°19′14.4″ N 21°54′42.9″ E), were selected for the research. The examined soils were classified on the granulometric composition basis as sandy loam—F1 and F2, and sand—F3 (according to the USDA classification). The exact characteristics of this area and detailed comparison of the three selected fluvisol are presented in Furtak et al. 2019 [26]. A map of the locations from which soil samples and water were taken is provided in Appendix A.

### 2.2. The Microcosm Experiment

For the microcosm experiment, 9 soil blocks (30 × 30 × 25 cm; 3 per each fluvisol: F1, F2, F3) including vegetation were collected in August 2018. Each of the blocks was transferred into a separate transparent, polypropylene container (33 × 33 × 42 cm) and flooded by water taken from the Vistula River above the place of soil sampling (Janowiec, Lubelskie Voivodeship; 51°19′06.8″ N and 21°54′53.5″ E) (Figure 2). The level of the floodwater above soil was 5 cm (Figure 2B). The containers were stored under 16 h daylight photoperiod, air humidity 55% and temperature of 20 °C. The water parameters and details of the microcosm experiment are presented in Furtak et al. 2020 [27].

To drain the water, drains were made in the containers, and the water was allowed to drain freely (Figure 2C). Soil samples were taken 7 days after flood condition and 28 and 56 days after the flood had receded.

### 2.3. The Natural Hydrological Event

In May 2019, in the areas from which the fluvisol used in the microcosm experiment were obtained, i.e., Janowiec and Wojszyn, there was an outflow of water from the riverbed (Figure 3). This situation was caused by abundant and intense precipitation in southern Poland [28]. Water persisted for 7 days after flooding. Samples were taken analogously to the control samples (Section 2.1.) 24 h after the water receded, i.e., when it was possible to reach the investigated soils. Soil samples were also taken 28 and 56 days after the flood had receded.

### 2.4. Soil Sampling

Each fluvisol’s control sample was a soil collected directly from site in 2018 (Figure 3 and Appendix A). Samples were collected with a probe from 0–20 cm depth from 15 representative spots (from an area of approximately 0.04 ha) per fluvisol and pooled (~1 kg). Soil samples were sieved through a 2 mm sieve (n = 3). 

Samples from the model experiment were taken after 7 days of flooding and after 28 and 56 days after the flood had receded (Figure 2A). From 3 containers containing the same fluvisol (n = 3 for 1 soil), 10 samples were taken randomly from a depth of 0–20 cm and then pooled (~0.5 kg). This yielded 3 samples for each fluvisol (n = 9). The samples were not sieved or dried. 

Samples from the natural flooding areas were collected after 7 days of inundation (Figure 3) and 28 and 56 days after the stress conditions had ceased. Samples were collected with a probe from 0–20 cm depth from 15 representative spots (from an area of approximately 0.04 ha) per fluvisol and pooled (~1 kg). Soils were not sieved or dried (n = 3).

All soil samples (n = 15) were quickly stored at 4 °C until analysis of the soil biological properties and metabolic potential of the microbial community. Table 1 presents the samples used in this research. The soil texture is presented in Appendix A.

### 2.5. pH Values and Moisture Contents

The soil pH in H_2_O was measured potentiometrically (1:2.5; 96 mV) after 24 h of incubation soil in sterile water (1:1) at room temperature (21 °C) (edge^®^ Multiparameter pH meter, HANNA Instrument, Woonsocket, RI, USA) [29]. Soil moistures were measured using the gravimetric (drying) method [30] and are presented in Appendix A.

### 2.6. Enzymatic Activity of Soil

The activity of soil dehydrogenases (DHa) was determined spectrophotometrically with 2,3,5–triphenyl-tetrazolium chloride (TTC) as a substrate [31]. The activities of DHa were reported as µg of triphenyl formazan (TPF) per g dry mass (d.m.) soil per incubation time of 24 h (Spectrophotometer UV-Vis Evolution™ 60, Thermo Fisher Scientific, Waltham, MA, USA).

The activity of alkaline (AlP) and acid (AcP) phosphatases were measured using ρ-nitrophenyl phosphate (ρ-NPP) as a substrate [32]. The analysis was performed using a spectrophotometer (Spectrophotometer UV-Vis Evolution™ 60, Thermo Fisher Scientific, Waltham, MA, USA), and results were reported as µg of ρ-nitrophenol (PNP) per g dry mass (d.m.) soil per incubation time of 1 h.

Enzymatic activity analyses were performed in triplicate for each soil sample.

### 2.7. Community Level Physiological Profiling (CLPP)

The CLPP method is often used to determine the influence of different environmental factors on the biological status of individual soil sites by tracking catabolic traits [33]. An assessment of the diversity of metabolic profiles of soil microbial populations reflects their state of activity [34]. The analysis is based on the use of a 96-well plate containing 31 different carbon sources in 3 replications, control and redox pigment-tetrazolium violet. Inoculum preparation involves the suspension of 1 g of fresh soil sample in 99 mL of sterile water, shaken for 20 min and incubated at 4 °C for 30 min [35]. Next, each of the 96 wells of the microplate was inoculated with 120 µL of soil inocula. All EcoPlates™ were incubated at 25 °C for 120 h. 

The intensity of the wells’ colour development was determined spectrophotometrically at λ = 590 nm [36] for a period 120 h at 24-intervals using a MicroStation ID (Biolog Inc., Hayward, CA, USA) plate reader at OD_590_.

The most intensive metabolic activity was observed after 120 h of incubation, and the results are presented in manuscript. The classification of substrates into 5 biochemical groups was made according to Weber and Legge (2009) [35].

### 2.8. Statistical Analysis and Visualization 

Statistical analyses were performed using Statistica ver. 10.0 (StatSoft. Inc., Tulsa, OK, USA). Significant differences were calculated according to post hoc Tukey’s HSD (Tukey’s honest significant difference) test, with a significance level of *p* < 0.05. Diagrams were performed using MS Excel software (Microsoft Corporation, 2016).

Microbial response in each EcoPlate™ that expressed average well colour development (AWCD) index was) calculated. AWCD is the primary indicator calculated from absorbance readings on EcoPlate™ plates. It is calculated based on the formula proposed by Garland and Mills (1991) [37,38,39,40].

Data measured as OD_590_ (optical density at λ = 590 nm) from utilization of each carbon source were visualised as heatmaps. Each cell’s colour indicates the value of the OD_590_ in the corresponding scale, from the minimum (blue) to the maximum values (red). Heatmaps were generated in R software (3.6.0) [41] using the *pheatmap2* (v. 1.0.12) package [42] applying UPGMA clustering. Box plots were generated using *ggplot2* (v. 3.3.2) package [43].

Selected results were also submitted to principal component analysis (PCA), which is a multivariate technique used to describe the relationship between several variables (e.g., soil chemical properties, enzymatic activity) and to explain the total variation in the data (Statistica ver. 10.0, StatSoft. Inc., Tulsa, OK, USA). PCA is a technique for reducing the dimensionality of such datasets, increasing their interpretability but minimizing the loss of information [44].

## 3. Results

### 3.1. Effect of Simulated and Natural Flooding on Soil pH

The pH values varied according to the soil type, the sampling date and the variant of the experiment (Figure 4). The greatest differences were recorded for F3.

In F1, after 7 days, there was an apparent decrease in pH value due to both NC and ME compared to the CS, but these differences were not statistically significant (Figure 4). After 28 days from the receding of flooding, a significant increase in pH was recorded in ME (at *p* < 0.05). Under NC, no difference was recorded. At 56 days, after the flood receded, pH values decreased in both experimental variants compared to values from 28 days. 

In F2, the pH values under ME conditions were decreased as a result of flooding compared to the CS, then increased 28 days after the flood receded and again decreased 56 days after flooding stopped (Figure 4). All these differences in pH values are statistically significant (at *p* < 0.05). Under NC, a decrease in pH values was recorded as a result of flooding, but no change was observed 28 days after the flood had receded. However, after 56 days, a significant decrease in pH values were observed.

In F3, a statistically significant decrease in pH was observed after 7 days of simulated flooding (at *p* < 0.05), an even greater decrease after 28 days of drainage and an increase after 56 days (Figure 4). Under NC, there was no change in pH values after 7 days of flooding, a decrease was seen after drying, but values after 28 and 56 days were similar.

Similarly to pH values, the enzymatic activity differed depending on the soil, stage and variant of the experiment. 

Comparing DHa activity in fluvisol in ME and NC (Figure 5), it could be seen that in all fluvisol the highest values were recorded after 7 days of flooding in both variants, but in NC it was statistically significantly lower (at *p* < 0.05). In F1 and F3 in both variants, there was a subsequent decrease in DHa activity during soil drainage. However, in F2 in ME after 56 days of drying, DHa activity was lower than in NC, but this was not statistically significant (at *p* < 0.05).

The overall trend of DHa activity in both NC and ME in all soils was as follows: increase after 7 days of waterlogging (relative to CS), decrease after 28 and 56 days of flooding.

Comparing AcP activity in soil in ME and NC, it could be seen that the highest values were recorded in F1 (Figure 6). There was a sequential decrease-increase-decrease in ME and an increase-decrease-increase in NC. With the ME conditions, these differences were statistically significant (at *p* < 0.05). In F2: decrease-increase-decrease in ME variant, and in NC increase-decrease-increase. However, these were apparent changes—not statistically significant in both variants of the experiment (at *p* < 0.05). In F3 in ME: decrease-increase-decrease, and in NC increase-decrease-decrease. The changes between AcP activity after 7 days of flooding and 28 after soil drying were statistically significant in both variants of the experiment (at *p* < 0.05).

Comparing the AlP activity in soil in ME and NC, it could be seen that the highest values were recorded in F1 (Figure 7). Comparing NC and ME conditions in F1 there was a statistically significant difference (at *p* < 0.05) only after 7 days of flooding, when it was higher in NC. In F2, the values obtained after 7 days of flooding were statistically significant and were higher in NC (at *p* < 0.05). In F3, AlP activity differed significantly after 28 and 56 days of soil drying, being significantly higher in the ME variant (at *p* < 0.05).

Based on the biplot (Figure 8), it can be concluded that the correlations between selected parameters and soils from different stages of the experiment varied from one fluvisol to another. 

For F1, samples from NC after 28 and 56 days of drying (NC_28, NC_56), as well as control soil (CS) and soil from ME after 56 days of drying (ME_56) were positively correlated with each other (Figure 8a). Interestingly, with this group of samples, DHa was negatively correlated, while AlP and AcP were positively correlated. The second group consisted of samples from ME after 28 days of drying (ME_28) and from NC and ME after 7 days of flooding (ME_7, NC_7). In addition, pH was negatively correlated with these samples. 

In F2, soil from ME after 7 days of flooding (ME_7) and from NC after 28 days of drying (NC_28) were positively correlated with each other (Figure 8b). DHa was also positively correlated with these samples, while AcP and pH were negatively correlated. The second group of samples were the control soil (CS), soils from ME after 28 and 56 days of drying (ME_28, ME_56) and soils from NC after 7 days of flooding (NC_7) and 56 days of drying (NC_56). AlP was positively correlated with them.

In F3, control soil (CS), soil from NC after 56 days of drying (NC_56) and soil from ME after 28 and 56 days of drying (ME_28, ME_56) were positively correlated (Figure 8c). AlP was also correlated with this group of samples, and pH was negatively correlated. The second group of positively correlated samples were soils from NC after 7 days of inundation (NC_7) and after 28 days of drying (NC_28) and from ME after 7 days of inundation (ME_7). DHa was positively correlated with this group of samples, and AcP was negatively correlated. 

### 3.2. Metabolic Potential of Soil Microorganisms under Simulated and Natural Flooding

Analysing the metabolic potential of microorganisms by the EcoPlate™ method based on the AWCD, it can be observed that, irrespective of the soil and the experimental variant, the highest values were found in fresh soils (Figure 9A). In F1 and F2, there was a decrease in AWCD values after 7 days of flooding, an increase after 28 days of drying and again a decrease after 56 days of drying under both NC and ME conditions. In F3 under ME there was recorded a decrease in AWCD after 7 days of flood and an increase after 28 and 56 days of drying. In contrast, under NC, a decrease in AWCD values was recorded at each stage of the experiment.

The analysis of the utilization of the different substrate groups shows that the utilization of amino acids under NC and ME conditions was similar in F1 and F2 after 7 days of feeding and 28 days of drying (Figure 9B), as well as in F1 also after 56 days of drying. In F2, a decrease in amino acid utilization was observed after 56 days of desiccation in ME, while an increase was observed in NC. The results obtained for F3 are similar between ME and NC, only, at the stage of 7 days of soil flooding, a much greater decrease in amino acids’ utilization was observed in ME compared to NC. 

Comparing the utilization of amines and amides under ME and NC conditions, we can observe that in F1 and F2 the trend was the same: decrease-increase-decrease (Figure 9C). In F3, on the other hand, there was a decrease-increase-increase in both ME and NC conditions.

The utilization of carbohydrates followed the same pattern as amines and amides (Figure 9D): decrease-increase-decrease in both variants of the experiment in F1 and F2; decrease-increase-decrease in the ME variant in F3; and decrease-decrease-increase in NC.

For the utilization of substrates from the carbohydrates group, the trend of decrease-increase-decrease was recorded in both ME and NC in all the fluvisol (Figure 9E).

Utilization of polymers followed the same pattern in NC and ME for F1 (Figure 9F). In F2, a difference was noted after 56 days of drying—in ME, utilization of polymers decreased at this stage, and in NC, it increased compared to the previous measurement. In F3, there was also a difference between ME and NC after 56 days of drying, but in ME, the utilization of polymers increased, and in NC, it decreased.

Based on the differences in the metabolic potential of the microorganisms (EcoPlate™) in the samples, a dendrogram was created (UPGMA, Euclidean distance matrix calculation method) (Figure 10). Results showed that microorganisms from all the fluvisol degraded α-Keto-Butyric Acid and D,L-α-Glycerol-Phosphate to a low degree. Another group of less degraded substrates included 2-hydroxy-Benzoic Acid, D-Malic Acid, α-D-Glucose-1-Phosphate, Putrescine, L-Threonine, m-Erythritol, β-Methyl-D-Glucoside, Gly-Glu. Clustering clearly distinguished samples F1 from all stages and variants and sample F2 after 7 days of flooding in natural (F2_NC_7) from the other samples. On average, the most intensive substrate utilization was found in F3, especially in the control (F3_CS) and from natural conditions (F3_NC) samples, which was visualised by the dark red colour of the cells on the heatmaps.

## 4. Discussion

### 4.1. Impact of Flooding on Soil Environment 

Soil water content mainly affects soil pH, diffusion of solvents and gases, or solubility of salts. It also impacts redox potential (Eh) and enzymatic activity [11]. It is also part of hydrolysis processes and determines the rate of carbon and nitrogen mineralization [7]. Already in 1984, Ponnampeuma observed that pH decreases as a result of flooding in alkaline soils [45]. On the other hand, in acidic soils, an increase in pH is observed as a result of changes in redox potential [46]. Furthermore, in the soil flooding microcosms, the authors observed higher pH as the incubation time progressed, which was significantly higher than the control at the later stage of incubation [47]. However, in soils before and after the 2010 flood (Vistula River) in Poland, researchers did not observe statistically significant differences in soil pH [48]. In ME and NC conditions, all the muds showed a decrease in pH at the time of flooding. However, after the flood conditions receded, the pH values arranged differently in F1, F2 and F3. These differences may be due to soil structure, which has a strong influence on both water contents and soil pH [49,50]. By this, in the F3, which was the lightest soil and contained the most sand (Appendix A), the pH changes were the most visible.

Enzymatic activity is a sensitive parameter that responds strongly to changes in the soil environment [51,52,53]. Soil moisture strongly affects soil enzymes [54]. The dehydrogenase activity determined in the present study was higher after flooding (day 7 of the flood) compared to fresh soils, under both ME and NC conditions in all the mudflats. These results are in agreement with those of other researchers who reported an increase of up to 90% of DHa activity in flooded soil and a rapid increase in DHa in flooded rice fields [10,55,56]. After 28 and 56 days after the cessation of flooding in both variants, DHa activity decreased in all the swards. This may indicate that DHa activity may have returned to its preflood stress state. This may be due to renewed access to oxygen in the soil, which is associated with the activity of aerobic organisms, and studies indicate that most soil dehydrogenases are produced by anaerobes [57]. Phosphatase activity is strongly correlated with pH [58], which is significant when flooding and changes in pH occur. Phosphatase activity after flooding in the present study varied both between the experimental variants and between the muds. In the case of AlP, a decrease after flooding was observed in ME in all muds and in F3 in NC, which is due to a decrease in pH [58,59]. In contrast, surprisingly, an increase in alkaline phosphatase after flooding was observed in F1 and F2 in NC. After the cessation of flooding in ME, i.e., at the time of oxygen emergence and at the decrease of humidity, an increase in AcP activity in ME was observed, which was accompanied by an increase in pH of these soils. In contrast, AlP decreased in NC after flooding had ceased, but this was also associated with a further decrease in pH. The rather high AlP activity in F1 and F2 may be due to the accumulation of phosphatase released into the soil by dying soil organisms and the reactivation of previously accumulated phosphatase [60]. These soils were thickened with vegetation that had died as a result of flooding, and phosphatases in addition to being of microbial origin are produced specifically in plant roots [51,60,61]. AcP activity did not differ after flooding in F2 and F3 under ME conditions because soil pH was also low. Low moisture influences a decrease in acid phosphatase activity by up to 31–40% [62], which is evident in the low AcP activity in the F3 fluvisol, which dried out quickly due to its structure. In the case of AlP in NC, a decrease in activity was observed in all muds 28 days after flooding had ceased, which is surprising because at this date the pH of the soils was still lower (7.22–7.49) compared to the control and flooded samples. In F1 and F2, 56 days after water receded, the pH dropped even further (7.15 and 7.20, respectively), and this was accompanied by an increase in activity, a result in line with expectations. The return of enzymatic activity (extracellular enzymes) to the preflood state was also observed by Schnecker et al. (2021) in a study on agricultural soils [63].

The results obtained from EcoPlate™ analysis are a valuable source of information regarding soil microbial activity [64]; however, due to the limited amount of literature data, it is difficult to discuss them. The utilisation of soil organic matter (SOM) by soil microorganisms is sensitive to changes in soil moisture content [65]. Studies using microcosms conducted to assess SOM decomposition and priming effects following flooding [66] showed differences in bacterial and archaeal species composition at two soil depths and changes in C-fixing capacity following flooding. The authors suggest that microorganisms switched from extracellular enzyme production to direct carbon incorporation and niches adapted to flooding conditions. The present study observed a decrease in metabolic activity (AWCD) following flooding (7 days) in ME, which may be related to a change in the microbiome’s own mode of C fixation [67]. In this study, we observed a decrease in the metabolic activity of soil microorganisms (expressed in the AWCD index), as a result of water stagnation, in all the muds and in both variants of the experiment. This may be related to a decrease in soil microorganism biomass because flooding decreases oxygen diffusion [8,68], and obligatory aerobic microorganisms die [69]. Furthermore, as a result of oxygen depletion, the abundance of Gram-negative bacteria and fungi naturally resident in well-aerated soil layers decreases [23,70,71]. Unger et al. (2009) observed a significant decrease in microbial biomass as a result of simulated flood conditions in a greenhouse [23]. However, the researchers acknowledged that the observations were too short to sufficiently estimate how much impact flooding has on changes in the soil microbial community. We concur with this position, as a single analysis of the impact of a stress factor on the soil environment provides little information. In addition, Liu et al. (2021) observed that, in the microcosms throughout the incubation process, microbial biomass carbon and nitrogen contents showed a decreasing trend under flooded conditions [47]. The authors also found that flooding altered the metabolic potential of the wetland soil microbial community. Furthermore, the functional diversity indices calculated by the researchers for all flood treatments were significantly lower than the control treatment after both 21 days and 132 days of incubation. In this experiment, 28 days after the flood receded in NC, a slow increase of AWCD was observed in F1 and F2. This may indicate changes occurring in the oxidoreduction potential. This is because, as a result of oxygen depletion, anaerobic microorganisms develop, which carry out numerous reduction and fermentation processes [72] and are therefore metabolically active. Among the anaerobic soil microorganisms, one may mention purple bacteria conducting anaerobic photosynthesis (*Rhodospirillum* sp.), sulphate-reducing bacteria (*Desulfoibrio* sp., *Desulfotoaculum* sp.) and nitrates (*Clostridium* sp.) and representatives of the Archea (archaeons), which produce methane as a result of respiration (*Methanobacterium* sp.) [73]. Using the results of the EcoPlate™ Biolog, we can observe that 28 days after flooding in the ME, the consumption of substrates from the groups amino acids, polymers, carboxylic acids, carbohydrates, amines and amides increased significantly. An increase in the relative utilisation rates of carboxylic acids and amino acids under flooding conditions by soil microorganisms was also observed by Liu et al. (2021) [47]. This may indicate the appearance of specialised microorganisms, which conduct anaerobic catabolism such as fermentation, acetogenesis and denitrification [74]. Moreover, plant residues remained in the containers (rotted as a result of water stagnation), which provided a large biomass load for the microorganisms. In NC, these changes are not so apparent, but it can be observed that the consumption of polymers and amines and amides in F1 and F2 increased after flooding receded. Among the microorganisms that degrade polymers are the anaerobic bacteria *Clostridium thermocellum* [75], the genus *Bacillus* spp. [76] and *Fibrobacter* spp. [77].

### 4.2. Microcosm vs. Natural Condition

Model studies have both advantages and disadvantages. Eller et al. (2005) and Kamplicher et al. (2001) claim that it is possible to extrapolate data from microcosms in relation to microbial community structure [19,78]. The researchers stress, however, that the reliability of such data depends on the scale and artificiality of the model experiments and the choice of experimental conditions. In the 1990s, a group of researchers opposed the widespread use of microcosms in ecological studies. Carpenter (1996) believed that without the context of appropriately scaled field studies, microcosm experiments become irrelevant and diversionary and have significant limitations [18]. Schindler (1998) also believed that smaller scale (micro- and mesocosm) experiments often yield erroneous conclusions about community and ecosystem processes [21]. Some researchers assume that the greater the diversity of species and environmental parameters in a model ecosystem, then the more difficult it is to maintain identical and stable conditions between repeated experiments [79]. Consequently, the more they will differ compared to natural conditions. The soil environment is very complex [80,81]; even at the microcosm scale, we have to deal with numerous species of microorganisms, worms, root exudates, soil structure, temperature, pressure and soil processes taking place, and then there is the factor under study, which, in the case of this study, is excessive moisture. Carpenter (1996) already stressed that some processes in ecosystems are too large, widespread or slow, and it is not possible to account for them in laboratory experiments (e.g., microorganism metabolism; nutrient cycling) [18]. However, he himself noted that properly designed, long-duration experiments can produce informative contrasts that test hypotheses and quantify effects at multiple scales simultaneously. An interesting application of microcosms is constructed wetland microcosms (CWMs), which are used to treat almost all types of wastewater using interactions between the substrate, macrophytes and microorganisms [82]. They are widely regarded as ecological and sustainable and are inexpensive and do not damage the ecosystem.

In this study for F1 and F3, we see that pH changes had a similar trend in ME and NC in terms of changes over time. In contrast, in F2 the difference appears at 28 days after the flood receded as pH increased significantly in ME, while in NC it remained at a similar level as before. DHa activity was significantly higher in ME conditions than in NC in all fluvisol. Furthermore, in F1 and F3, we observed that DHa arranged itself similarly in ME and NC during the experiment, while in F2 the NC differed significantly from ME after flooding had ceased. In contrast, the changes in activity of both phosphatases during the experiment were significantly different between ME and NC. In the case of AWCD, values followed a similar pattern only in F1. In F2 and F3, differences were observed between the course of changes in ME and NC in samples collected after the cessation of flooding. There may be many reasons for such differences between ME and NC. Soil is a complex environment, which affects and limits the microcosms pathway. In the flood experiment, we first deprived the object of the possibility of natural, free leaching of water. Under natural conditions, there are many layers of soil and groundwater beneath them. Floodwater is successively absorbed and transported deep into the soil. In the model experiment, we had a closed object, and as long as the holes were plugged, the water had no chance to drain away. Under laboratory conditions, we excluded the influence of climate–solar radiation and wind. Although the temperature was controlled, wind and radiation heating of the soil surface would accelerate evaporation [83], causing faster water depletion. The limited number of plants in the micro- and mesocosm also played a role. Under natural conditions, these areas contain shrubs and trees whose root systems take up more water than grassy plants [84]. Moreover, enzymatic activity is a very sensitive and variable parameter [51,61,85,86]. The presence of plant residues or dead soil organisms in the studied soil affects their activity and the biomass of microorganisms [87,88]. In the case of floods, such residues are found in the soil, and by sampling a small area of microcosms, we have a smaller number of residues. The community of microorganisms in the microcosm study was also spatially limited. The containers did not include representatives of the entire soil microbiota due to the fact that some of them reside at specific soil depths [89,90] or in the rhizosphere of specific plants [91,92], yet in such a complex environment the interactions between these microorganisms are important. A study by Höhener et al. (2006) on biodegradation of hydrocarbon vapours showed that all field models suggested a significantly higher rate of benzene degradation than that measured in the laboratory, which, according to the authors, suggests that the field microbial community was better at developing benzene degradation activity [93]. This may be due precisely to the interactions between microorganisms and each other and between plants and other soil organisms, which are limited by the laboratory container. In the microcosm, we also disturb the stability between predators and prey, both in relation to bacteria/fungi and protozoa. Prey are limited by the amount of available nutrients, and once the prey have been depleted, the predators are also starved [94]. A disturbance in the microbial population (abundance and diversity) is evident in their metabolic activity, as determined by EcoPlate™ and in the enzymatic activity of the soil.

Researchers who have managed to compare results obtained from model experiments and natural conditions (field experiments) have often noted differences between their models and natural conditions. In a study on the effect of Cu on indigenous soil microorganisms, Ranjard et al. (2006) observed differences between the response of microorganisms in the microcosm and the field experiment [95]. No significant changes in biomass C were observed in the field study compared to the model experiment. The authors concluded that, in the field experiment, the effects of Cu contamination may be overcome or hidden by pedoclimatic (temperature, water content and aeration in soil) changes. Interestingly, similar studies on the effect of metals on the microbial community have shown a smaller effect of contamination in microcosm experiments compared to field plots [96]. In the literature, it could be also found results that show the comparability of data obtained from model experiments and field experiments. In 1992, Teuben and Verhoef, comparing data obtained from micro- and mesocosm with data obtained from the forest floor, observed the same trend in all variants of the experiment [97]. Burrows and Edwards (2004) investigated the effect of carbendazim on the soil environment (e.g., dehydrogenase activity, soil ammonium-N and nitrate-N concentrations, microorganism biomass) using an integrated soil microcosm (ISM) test protocol (i.e., a small soil microcosm with vegetation and earthworms) [98]. By comparing their results with data from other small microcosms, large model terrestrial ecosystems and field studies, the researchers concluded that soil microcosm testing can adequately predict environmental effects. In an experiment on trace element enrichment in field crops following the addition of fly ash, the researchers showed that microcosms accurately predicted the enrichment ratios (ER) of 22 of 25 trace elements analysed in field-grown lucerne [99].

Short-term experiments with stable incubation conditions in the laboratory allow the assessment of the potential impact of different factors on the soil microbiome. Benton et al. (2007) highlight the face that those small-scale experiments using “model organisms” in microcosms or mesocosms can be a useful approach to seemingly unsolvable global problems, such as ecosystem and these responses to climate change or biodiversity management [100]. The researchers point out that such experiments can be combined with theory development and act as a stimulus for further and larger studies. However, one must be careful when collating microcosm and field data from different environments. Extrapolation should only apply to carefully planned ecosystems involving the same soil and vegetation. Further work is needed to integrate and coordinate microcosms and field experiments to investigate how scale, type of stressor applied, soil type, climate and other environmental factors influence the results obtained from laboratory experiments.

## 5. Conclusions

The present study showed that flooding under both natural and laboratory conditions affects soil pH, enzymatic activity and metabolic potential of soil microorganisms. The changes of these parameters are more pronounced in microcosm experiment than in field conditions. The trend of pH value changes was the same under natural and model conditions for all soils; dehydrogenase activity changes were the same for medium (F1) and light (F2) fluvisol; phosphatases activities were different. The metabolic activity potential (AWCD) had the same trend in both variants, only in the medium fluvisol (F1). AWCD index in light (F2) and very light fluvisol (F3) had the same decreasing trend only in the first sampling date, i.e., after 7 days of flooding. Drying had different effects on these parameters under natural and laboratory conditions. Depending on the soil (its type, structure, vegetation), some parameters could return to the state before the stress conditions. 

Could model experiments provide insight into the response of the soil environment to flooding? Although differences between results obtained from microcosm experiment and natural condition have been reported, we believe that model studies are needed. When analysing the effects of flooding on the biological activity of soils, they allow information to be obtained on changes in basic soil processes (pH, dehydrogenases activity, metabolic potential associated with carbon use). At the same time, it is not possible for researchers to plan and carry out comprehensive studies on flooding under natural conditions, because an extreme event flooding is difficult to predict and the estimation of the area where control (preflood) samples should be taken is rarely possible. The soil flood microcosms allow observation of the soil environment under water stress conditions, and this is important from the point of view of climate change and increasing frequency of hydrological events (i.e., flooding) worldwide. Future microcosm studies of flooding soils should consider using larger containers so that more soil volume and depth are available. It is also worth including more parameters in the study—i.e., oxygen content, activity of other enzymes (e.g., catalase)—or assessing the presence of fermenting microorganisms.

## Figures and Tables

**Figure 1 biology-11-00386-f001:**
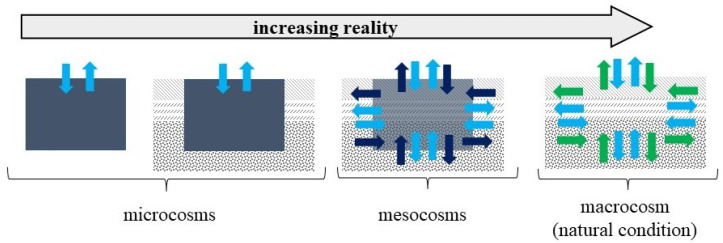
Microcosms, mesocosms and macrocosm in soil analysis. Blue arrows—exchange of components (e.g., O_2_, CO_2_, water, nutrients); dark blue arrows—controlled movement of soil organisms; green arrows—unrestricted movement of soil organisms. Author’s elaboration based on [19].

**Figure 2 biology-11-00386-f002:**
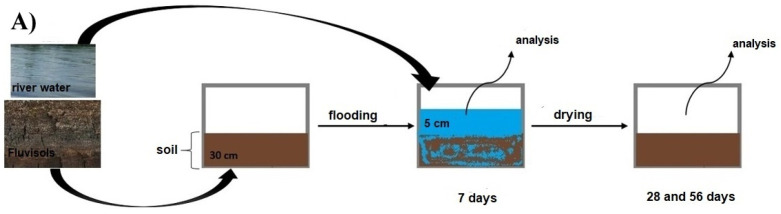
(**A**) Experiment scheme; (**B**) containers with flooded soil (7 days of flooding); (**C**) containers with drained soil (56 days after the floods ended) (author’s materials).

**Figure 3 biology-11-00386-f003:**
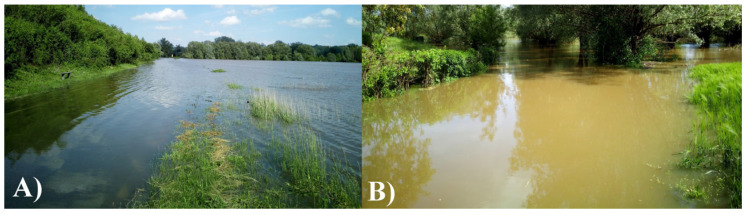
Photos of areas flooded by waters (7 days) from the Vistula River: (**A**) Janowiec; (**B**) Wojszyn (author’s materials, unpublished).

**Figure 4 biology-11-00386-f004:**
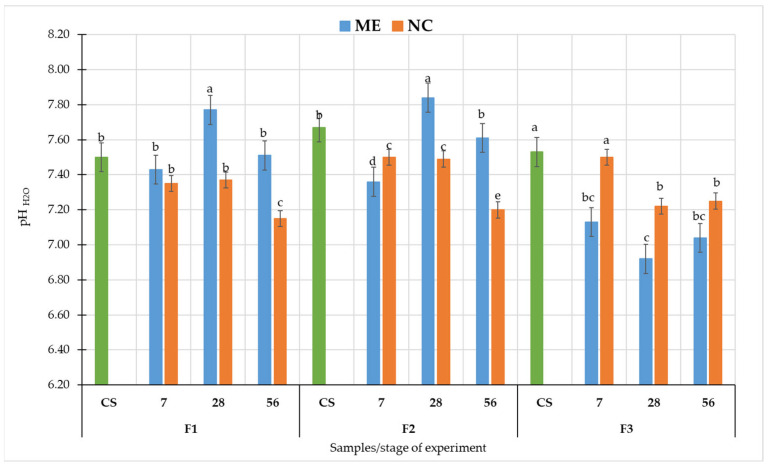
Soil pH_H2O_ values in different experimental conditions. Different lower-case letters “a–e” indicate significant differences for the same soil sample in different experimental variants (control, simulated flooding, natural flooding) at *p* < 0.05 (Tukey HSD test; n = 9). Explanation of the samples in Table 1.

**Figure 5 biology-11-00386-f005:**
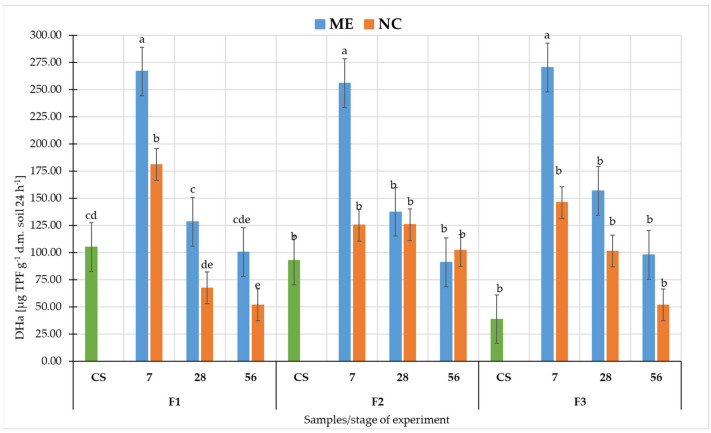
Activity of dehydrogenases (DHa) in fluvisol. Different lower-case letters “a–e” indicate significant differences for the same soil sample in different experimental variants (control, simulated flooding, natural flooding) at *p* < 0.05 (Tukey HSD test; n = 9). Explanation of the samples in Table 1.

**Figure 6 biology-11-00386-f006:**
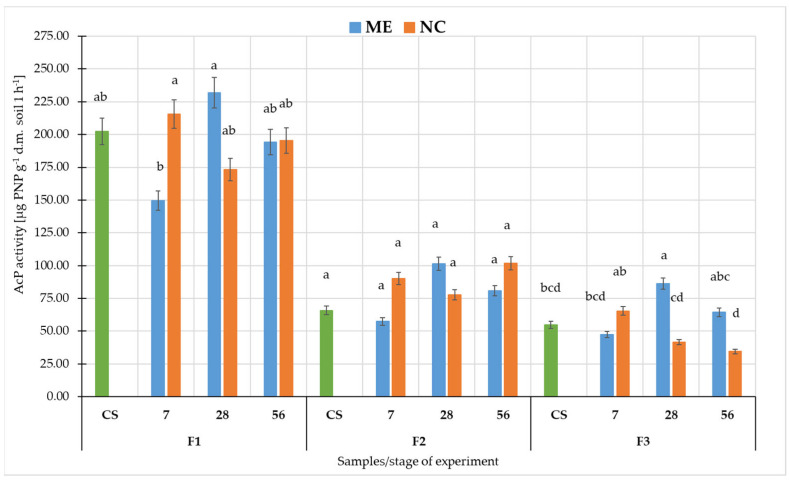
Activity of acid phosphatases (AcP) in fluvisol. Different lower-case letters “a–d” indicate significant differences for the same soil sample in different experimental variants (control, simulated flooding, natural flooding) at *p* < 0.05 (Tukey HSD test; n = 9). Explanation of the samples in Table 1.

**Figure 7 biology-11-00386-f007:**
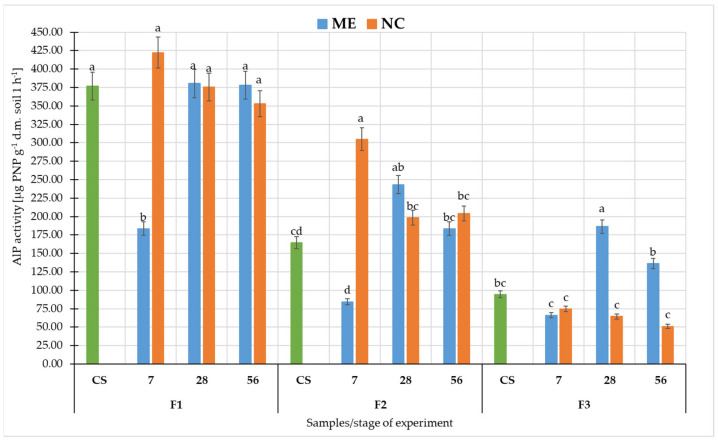
Activity of alkaline phosphatases (AlP) in fluvisol. Different lowercase letters “a–d” indicate significant differences for the same soil sample in different experimental variants (control, simulated flooding, natural flooding) at *p* < 0.05 (Tukey HSD test; n = 9). Explanation of the samples in Table 1.

**Figure 8 biology-11-00386-f008:**
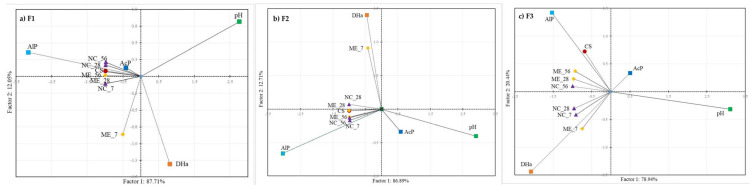
Principal component analysis (PCA) between dehydrogenase (DHa), acid phosphatase (AcP), alkaline phosphatase (AlP) activities, pH values and soil samples in different experimental stages: (**a**) F1; (**b**) F2; (**c**) F3. Explanation of the samples in Table 1.

**Figure 9 biology-11-00386-f009:**
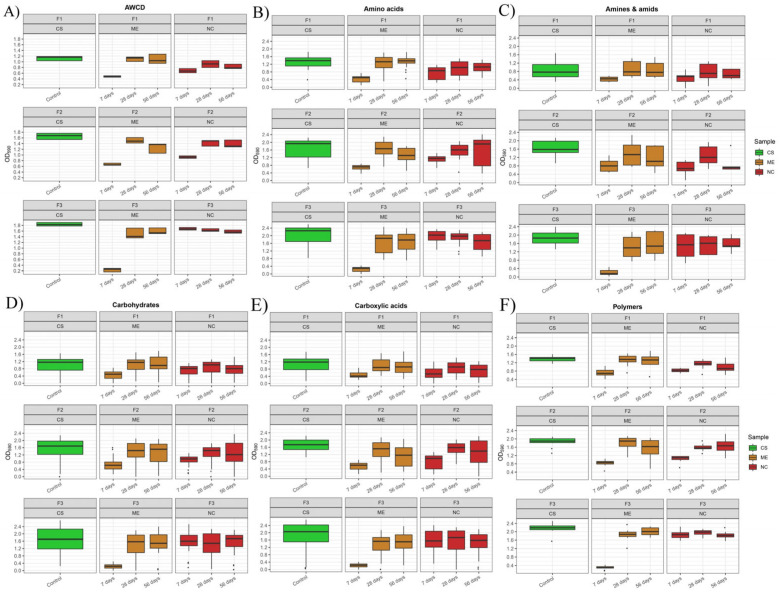
Change in the AWCD (average well colour development) and utilization of particular carbon substrate groups from EcoPlate™ by microorganisms during the model experiment and natural condition: (**A**) AWCD; (**B**) amino acids (n = 6); (**C**) amines and amides (n = 2); (**D**) carbohydrates (n = 10); (**E**) carboxylic acids (n = 9); (**F**) polymers (n = 4). Box plot: whiskers represent the minimum and maximum values, the horizontal line in the box indicates the median, points under the box indicate outliers. Explanation of the samples in Table 1.

**Figure 10 biology-11-00386-f010:**
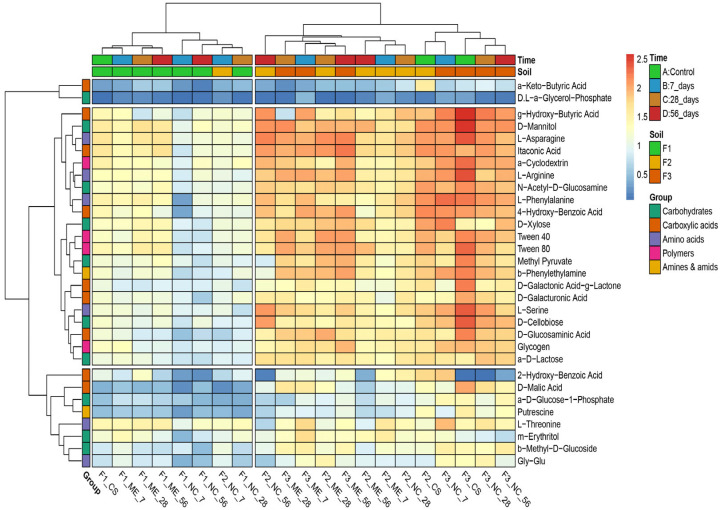
Heatmap and clustering tree for utilization of individual carbon sources (EcoPlate™ Biolog^®^) by the soil microorganism community from three fluvisol during the model experiment and natural condition. Lack of or low utilisation is represented by a dark blue colour; the gradient from light blue to red represents positive utilisation; the colour key scale (0.5–2.5) for each substrate is based on dye reduction, quantified by Omnilog units (EcoPlate™ Biolog^®^). Explanation of the samples in Table 1.

**Table 1 biology-11-00386-t001:** Samples used in the experiment—abbreviations, location, terms.

Abbreviation	Fluvisol	Location	Condition	Day of Flooding	Day of Drying
CS	F1	Wojszyn; 51°20′03.4″ N 21°56′43.2″ E	Control sample ^1^	-	-
ME			Microcosm experiment	7	28, 56
NC			Natural condition	7	28, 56
CS	F2	Janowiec; 51°19′29.9″ N 21°55′19.2″ E	Control sample ^1^	-	-
ME			Microcosm experiment	7	28, 56
NC			Natural condition	7	28, 56
CS	F3	Janowiec; 51°19′14.4″ N 21°54′42.9″ E	Control sample ^1^	-	-
ME			Microcosm experiment	7	28, 56
NC			Natural condition	7	28, 56

^1^ soil without water stress; fresh soil collected August 2018.

## Data Availability

Not applicable.

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
