# Peer review of "Can Model Experiments Give Insight into the Response of the Soil Environment to Flooding? A Comparison of Microcosm and Natural Event"

_biology, 2022, doi:10.3390/biology11030386_

Round 1

Reviewer 1 Report

The manuscript titled “Could model experiments provide insight into the response of the soil environment to flooding? Comparison of microcosm and natural event” is a work that can positively contribute to the field. I found no major revisions; however, the language mistakes, style, and grammar must be carefully revised. Here there are only a few examples:

Line 2

Could model experiments provide insight into the response of the soil environment to flooding? Comparison of microcosm and natural events.

Line 18

and the course must be carefully planned.

Line 27

In addition, depending on the soil (its type, structure, vegetation), some of the parameters may have

Line 62

They observed a significant decrease in the biomass of microorganisms.

Line 66

but the authors point out that this

Line 106

and details about the microcosm experiment are presented in Furtak et al. 2020 [21].

Line 125

The control sample for each fluvisol was soil collected directly

Line 158

Analyses of enzymatic activity were performed in triplicate on each soil sample.    

Line 200

In F2 the pH values in ME were decreased as a result of flooding compared

Line 271

In F2, the first group of positively correlated samples was soil

Line 495

The more, they will differ from natural conditions. REPHRASE

Author Response

Dear Reviewer,

Thank you very much for appreciating our work. And for your valuable comments. Corrections were made in the indicated places with tracking changes. We also provided linguistic correction.

Comment: Line 2. Could model experiments provide insight into the response of the soil environment to flooding? Comparison of microcosm and natural events.

Answer: Thank you for your comment. A sentence has been reworded.

Comment: Line 18. and the course must be carefully planned.

Answer: Thank you for your comment. A sentence has been reworded.

Comment: Line 27. In addition, depending on the soil (its type, structure, vegetation), some of the parameters may have

Answer: Thank you for your comment. A sentence has been reworded.

Comment: Line 62. They observed a significant decrease in the biomass of microorganisms.

Answer: Thank you for your comment. A sentence has been reworded.

Comment: Line 66. but the authors point out that this

Answer: Thank you for your comment. A sentence has been reworded.

Comment: Line 106. and details about the microcosm experiment are presented in Furtak et al. 2020 [21].

Answer: Thank you for your comment. A sentence has been reworded.

Comment: Line 125. The control sample for each fluvisol was soil collected directly

Answer: Thank you for your comment. A sentence has been reworded.

Comment: Line 158. Analyses of enzymatic activity were performed in triplicate on each soil sample.  

Answer: Thank you for your comment. A sentence has been reworded.

Comment: Line 200. In F2 the pH values in ME were decreased as a result of flooding compared

Answer: Thank you for your comment. A sentence has been reworded.

Comment: Line 271. In F2, the first group of positively correlated samples was soil

Answer: Thank you for your comment. A sentence has been reworded.

Comment: Line 495. The more, they will differ from natural conditions. REPHRASE

Answer: Thank you for your comment. A sentence has been reworded.

Reviewer 2 Report

You have provided a good argument to justify the use of microcosms to predict changes in soil conditions following inundation, but you should have discussed certain omissions in the conduct of the experiment such as absence of data on environmental conditions (e.g. length of storage, temperature, humidity, soil moisture levels, etc  as well as the exact positions of the sampling probes over the area of the microcosms). 

More importantly a major re-arrangement of the text is required. For instance, a large portion of chapter 4.2 consists of a review of the work of previous authors and so should be much abbreviated and transferred to the Introduction.

Also, the constant switching between results for ME and NC data in the Results and Discussion chapters render the text almost unreadable in parts. These sections should be re-designed to ease comprehension. For instance,  results for the various types of microcosm could be described separately where appropriate in both chapters, and whilst doing this, the text should be considerably abbreviated.

Finally there are many mistakes in spelling and grammar that a native English speaker should have easily corrected.

Please examine closely the many suggestions for improvement that I have highlighted on the attached returned pdf

Author Response

Dear Reviewer,

Thank you very much for the time you have devoted to our work and for your many valuable comments.

We have made corrections in all the places indicated in the PDF with tracking changes.

We will post our responses to your comments below.

Comment: You have provided a good argument to justify the use of microcosms to predict changes in soil conditions following inundation, but you should have discussed certain omissions in the conduct of the experiment such as absence of data on environmental conditions (e.g. length of storage, temperature, humidity, soil moisture levels, etc  as well as the exact positions of the sampling probes over the area of the microcosms). 

Answer: We added information on the experimental conditions in the laboratory at L128-132. We refer in text to a publication in which the experimental conditions were described in great detail L... Furtak et al. 2020 (https://doi.org/10.1016/j.catena.2019.104448). The reference was made in order not to duplicate the content (plagiarism).

Information about soil moisture during the experiment was placed in the supplement (Table S1), because due to the lack of repetition of this analysis we cannot perform statistical analysis and therefore we believe that these results should not appear in the main text.

Storage of samples - all analyses were performed immediately after sample collection, so they were stored briefly as written. Due to identical storage conditions for all samples we did not include details in the paper.

Comment: More importantly a major re-arrangement of the text is required. For instance, a large portion of chapter 4.2 consists of a review of the work of previous authors and so should be much abbreviated and transferred to the Introduction.

Answer: We have redrafted the discussion. Part of the chapter has been moved to the Introduction (L74-96). Part has been left in the discussion. A few passages have been deleted (L501-542; L575-576). We hope that this form will now be accepted. Further in 4.2, there is a bit of a literature review, but it is intended to give the reader an idea of the problem that there is a lack of data in the literature on the comparison of laboratory results with natural conditions.

Comment: Also, the constant switching between results for ME and NC data in the Results and Discussion chapters render the text almost unreadable in parts. These sections should be re-designed to ease comprehension. For instance,  results for the various types of microcosms could be described separately where appropriate in both chapters, and whilst doing this, the text should be considerably abbreviated.

Answer: We do not really understand the above comment. The purpose of the study is to compare ME and NC. To compare the results of both variants of the study and to check whether the results obtained from them are compatible. Hence, such switching between ME and NC in the results. There are no "various types of microcosms"; the experiment was one - flooding for 7 days, drying (of the same samples) for 28 and 56 days, in three repetitions for each soil. The experiment was conducted continuously. ME just stands for experimental conditions, while NC stands for natural conditions - a flooding that occurred naturally in the study areas. The paper is about comparing these two variants of flooding (simulated and natural), which is why there are so many switching between NC and ME in the text. It seems to us that describing these results separately, i.e. first ME alone, then NC alone misses the purpose of the paper. We wanted to compare these results with each other

Comment: Finally there are many mistakes in spelling and grammar that a native English speaker should have easily corrected.

Answer: We have taken care of the linguistic corrections. Thank you for your comment.

Comment: L22. metabolic potential – explain this term

Answer: “Metabolic potential” is the potential of microorganisms to decompose/metabolise individual carbon sources placed on the EcoPlate. Details are in the methodology L190-193. and in the referenced publications. We have added some information in the methodology. There is no place for this in the abstract. The phrase "metabolic potential" is commonly used in the context of research using EcoPlate™. Please take a look references:37-40, 35, 33, 34

Comment: L23 “obtained” delete

Answer: deleted as suggested.

Comment: Average well colour development – explain this term

Answer:  is an index (AWCD) which is calculated based on the colouration of the EcoPlate wells. There is a literature reference for it in the methodology explaining how it is calculated. Some info added though, but in the methodology L216-219. Not in the abstract.

Comment: L29 insert „are needed”

Answer: corrected as suggested

Comment: L37 change to „conducts”

Answer: corrected as suggested

Comment: L45 Change to 'height of storm waves'

Answer: corrected as suggested

Comment: L67 Change to 'flooding'

Answer: corrected as suggested

Comment: L99 - Give reference for this and provide a more detailed description of the soil types, eg. alluvial with X amounts of sand, gravel clay, humus etc

Answer:  All these details are in the referenced paper by Furtak et al. 2019 (doi:10.3390/su11143929) and Furtak et al. 2020 (https://doi.org/10.1016/j.catena.2019.104448); however, they are also included in the supplement as Table S1.

Comment: L71 Please explain these terms. In so doing, it would be helpful to place Figure 10 here.

Answer: clarified, moved figure 10 as further suggested

Comment: L75 Replace with 'are'

Answer: corrected as suggested

Comment: change to 'excludes'

Answer: corrected as suggested

Comment: L93 Change to 'soils classed as fluvisoils [19]'

Answer: corrected as suggested

Comment: L98 Give reference for this and provide a more detailed description of the soil types, eg. alluvial with X amounts of sand, gravel clay, humus etc

Answer: Zachęcamy do zapoznania się ze szczegółami w pracy Furtak et al. 2020; część informacji dodaliśmy do suplementu jako Table S1.

Comment:  L104: explain how water sample is 'above' place of sampling. Is the river at a higher level?

Answer: As stated in the referenced paper by Furtak et al. 2020 (https://doi.org/10.1016/j.catena.2019.104448) - water was taken above the sampling point, i.e. we went back upstream. We added a map as a supplement. Figure S1.

Comment: L101 change to 'For'

Answer: corrected as suggested

Comment: L101 insert comma

Answer: corrected as suggested

Comment: L104 insert comma

Answer: corrected as suggested

Comment: 1) State the environmental conditions within the laboratory (temperature and humidity regimes will affect drying and chemical/microbial activity).

Answer: All these details are in the referenced paper by Furtak et al. 2019 (doi:10.3390/su11143929).

Comment: 2) Were the sides of the containers exposed to sunlight? –

Answer: No, the samples were not exposed to sunlight. Illuminated with ordinary artificial light. 16-h day-light photoperiod, ambient air temperature of 20 °C.; Humidity - 55%.

Comment: L120; again, measurements of soil moisture levels should have been taken at these Times Answer: we added soil moisture measurements as a supplement - Table S1. Due to the lack of repetitions of this analysis we cannot perform a statistical analysis and therefore we believe that these results should not appear in the main text. Replications were not performed due to the limited amount of soil material.

Comment: L125 Here and elsewhere, change to 'fluvisol'

Answer: corrected as suggested

Comment 14: L126: “State quantity of soil per sample and whether the whole profile from 0-20 cm was collected.” –

Answer:–  Yes, the whole 0-20 cm was taken. Approximately 1 kg of soil was obtained for each sample, as further written in the text. It was not weighed how much by weight fits into a 20 cm high probe at a time. Nor have we ever encountered weighing such a single "prick" of soil.

Comment: L129 insert 'as per Fig 1 A'

Answer: corrected as suggested

Comment 15: L130: „receded” „'had been drained' 

Answer: the samples were not dried artificially, only holes were made in the containers and the water was allowed to drain off on its own. That is why the word "dried" is not used in the text. Details are in the referenced paper by Furtak et al. 2020 (https://doi.org/10.1016/j.catena.2019.104448) and L136-138.

Comment: L131: Samples should not have been taken completely randomly. Those near edges of containers will have experienced a different environment from those in the centre. If this was not taken into account when sampling, it should be discussed as a possible source of error amongst variables – Answer: samples were taken randomly, i.e. from both the edges and the centre, and then pooled and mixed as one overall sample to reflect changes throughout the container. Furthermore, in the literature on soil environmental research, pooling of samples is common, e.g.: https://doi.org/10.1007/s10967-019-06472-2, doi: 10.25081/jebt.2019.v11.5506, https://doi.org/10.1016/j.apsoil.2019.01.004, 10.1016/j.apsoil.2017.10.033.

Comment: L139: State how long samples were stored before analysis and whether the fluvisols were all stored for similar periods of time. (Some differences amongst variables could be influenced by length of storage).

Answer: As written, all samples were stored briefly at the same temperature and under the same conditions. In terms of timing, EcoPlate™ analysis and pH determination was performed the day after sampling, enzyme activity determination up to 2 days after sampling. The analyses were performed on an ongoing basis immediately after sampling, i.e. fresh in 2018, from the model experiment in 2018, and from the natural flood in 2019. Due to the fact that the samples were stored the same and for the same amount of time, this information was not included in the manuscript.

Comment: L142 Insert 'collected August 2018; the others collected May 2019. If storage times differ these should be included in the Table

Answer: This information has been added. No, storage times were not different. Samples taken in 2018 were analysed directly in 2018 and samples taken in 2019 were analysed on an ongoing basis in 2019.

Comment: L146: Why necessary to incubate for as long as 24 h and at what temperature?  If available, provide a reference for this procedure.

Answer: this is the standard method used in our laboratory. however, literature has been added (L174, ref. 29).

Comment: L150 Change to 'were'

Answer: corrected as suggested

Comment: L153 Change to 'were measured using'

Answer: corrected as suggested

Comment: L156 change to 'were'

Answer: corrected as suggested

Comment: L158 Change to 'sample'

Answer: corrected as suggested

Comment 19: L159: Community Level Physiological Profilling (CLPP)

Answer: Some information has been added (L190-204; 210-218). Reference has been made to other paper (ref. 33- 40). The method according to us has been described quite accurately. Please see: https://www.ncbi.nlm.nih.gov/pmc/articles/PMC3376570/

Comment 20: L161-162: 1 g of fresh soil sample was suspended in 99 mL of sterile water, shaken for 20 min. and incubated at 4 °C for 30 min. [25]. - Where is this coming from? \

Answer: I do not understand the remark. The methodological procedure is described and the references is given.

Comment: L164 change to 'inocula'

Answer: corrected as suggested

Comment 21: L179: Explain this term

Answer: We do not understand the comment. Heatmaps are a method of representing data graphically where values are depicted by color, making it easy to visualize complex data and understand it at a glance. Now its method is very common. Some information has been added: L216-220 and L381-386. Please see: 10.3390/f10121083; https://link.springer.com/article/10.1007/s00253-021-11449-x; https://doi.org/10.3390/f10121083.

Comment 22: L183: Explain this term

Answer: PCA - principal component analysis. It is commonly used for dimensionality reduction by projecting each data point onto only the first few principal components to obtain lower-dimensional data while preserving as much of the data's variation as possible. The first principal component can equivalently be defined as a direction that maximizes the variance of the projected data. PCA is the simplest of the true eigenvector-based multivariate analyses and is closely related to factor analysis. We have not encountered a detailed explanation of statistical methods in publications. This does not seem to us to be necessary to describe. Then it would be necessary to describe all the methods (ANOVA, Tukey etc.), whereas we think that the researchers are knowledgeable with them. e.g. doi:10.1007/b98835. Some information has been added: L223-229.

Comment: L188 Replace with 'soil type'

Answer: corrected as suggested

Comment: L191 This Figure is too complex. It could be simplified by  separating F1, F2 and F3 with more complete vertical lines and providing the colour system shown in Fig 4. Then check you have described results correctly in text! (see major mistake with description of Fig 5 below)

Answer: Figure 3 has been corrected as suggested.

Comment: L195 insert 'after 7 days'

Answer: corrected as suggested

Comment: L195 replace with 'an apparent'

Answer: corrected as suggested

Comment: L199 change to in NC, but not in ME

Answer: corrected as suggested

Comment: L200 Omit 'were'

Answer: corrected as suggested

Comment: L201 add 'd' here and elsewhere

Answer: corrected as suggested

Comment: L212 Insert '(Figure 4)'

Answer: corrected as suggested

Comment: L212 here an elsewhere use past tense

Answer: corrected as suggested

Comment: L217: Was it sig. at P < 0.05?

Answer: No, there was no difference. However, for clarity, the P in all figures was changed to P < 0.05. For example, below is a table with the results of the Tukey HSD test for DHa at P  < 0.01 and P < 0.05.

F1

DHa

P < 0.05

P < 0.01

CS

104.85

cd

cd

7 ME

266.63

a

a

7 NC

180.9

b

b

28 ME

128.17

c

bc

28 NC

67.48

de

cd

56 ME

100.38

cde

cd

56 NC

51.73

e

d

F2

DHa

P < 0.05

P < 0.01

CS

92.65

b

b

7 ME

255.89

a

a

7 NC

125.111

b

b

28 ME

137.33

b

b

28 NC

125.7

b

b

56 ME

90.99

b

b

56 NC

101.83

b

b

F3

DHa

P < 0.05

P < 0.01

CS

38.61

b

b

7 ME

270.35

a

a

7 NC

145.96

b

b

28 ME

156.63

b

b

28 NC

101.36

b

b

56 ME

97.73

b

b

56 NC

51.79

b

b

Comment: L224 insert 'and 56 days'

Answer: corrected as suggested

Comment: L225 Suggest replace with '28-56 days to levels close to CS'

Answer: corrected as suggested

Comment: L227-239 Please check this description of results. It does not appear to agree with the differences in significance indicated in Fig 5. Check this description. The increase/decrease sequences appear to be in the wrong order. Also it should be noted that most apparent differences were insignificant at P < 0.01. Why did you not show results for P < 0.05?

  1. I think you have confused this description with that for DHa!

Answer: description checked, corrected, shortened. It is now correct.

Comment: L256 Again this description does not fully tally with significance values cited in Fig 6. For instance, for F3 all NC differences are insignificant. You have not mentioned the clear differences amongst the three sites

Answer: description checked, corrected, shortened. It is now correct.

Comment: L262 Here and elsewhere where appropriate  change to 'fluvisol'

Answer: corrected as suggested

Comment: L316 Explain this term

Answer: error crept in, corrected

Comment: L316 This word is not understood. Choose another one.

Answer: corrected as suggested

Comment: L317, L318 Change to 'Utilization of polymers'

Answer: corrected as suggested

Comment: L323 Insert 'of'

Answer: added as suggested

Comment: L329 Briefly describe the use and interpretation of the 'heatmap'

Answer: An explanation has been added in figure caption.

Comment: L330 Spelling

Answer: corrected as suggested

Comment: L334 State how this was distinguished

Answer: An explanation has been added.

Comment: L335 should this be F3_NC?

Answer: yes; corrected

Comment: L340 Spelling

Answer: corrected as suggested

Comment: L347 replace with full stop

Answer: corrected as suggested

Comment: L349 delete

Answer: deleted as suggested

Comment: L350 Rephrase - meaning not clear.

Answer: corrected as suggested

Comment: L355 delete ,

Answer: deleted as suggested

Comment: L357 90% of what?

Answer: corrected as suggested

Comment: L365 delete ,

Answer: deleted as suggested

Comment: L379 Wrong word!

Answer: corrected as suggested

Comment: L384 activity of what?

Answer: corrected as suggested

Comment: L391 Define this term

Answer: An explanation has been added.

Comment: L392 Explain use of 'marsh'

Answer: corrected as suggested

Comment: L409 insert comma

Answer: corrected as suggested

Comment: L410 insert comma

Answer: corrected as suggested

Comment: L420: State how nitrates are being affected

Answer: We do not understand the remark: if we are to describe the effect of flooding and nitrifying microorganisms on nitrate, we should also describe the effect of other bacteria on individual parameters, and this is not what this paper is about.

Comment: L425 Do you mean 'amides'

Answer: yes; corrected as suggested

Comment: L432 spelling!

Answer: corrected as suggested

Comment: L433 insert 'are'

Answer: corrected as suggested

Comment: L435 rozdział 4.2. - Most of this chapter consists of a review of the literature. So, lines 436-511 should be considerably abbreviated and transferred to the Introduction.

Answer: yes, it is a literature review, but not an introduction to the subject, but a search for the opinion of other researchers in the context of using ME in environmental studies. I have reworded this passage a bit, but we think that the greater part of the text fits precisely into the discussion in which we want to find an answer to the question whether other researchers have obtained similar results to ours.

Comment: Descriptions in lines 511-521 should be rearranged to avoid continually switching between ME and NC results.

Answer: But that is what this paper is all about. It is about comparing ME and NC and assessing whether ME can correspond to NC. Describing this separately does not seem to fit with the purpose of the paper. We do not really understand the above comment. The purpose of the paper is to compare ME and NC. The aim of the paper is to compare ME and NC, to compare the results of both variants of the study and to check whether the results obtained from them are compatible. Hence, such jumps between ME and NC in the description of results. The description of the results has been slightly improved, but we do not see the point of describing separately ME and NC, because we only want to compare these results with each other.

Comment: L444 use another term, e.g. 'shone light on'

Answer: corrected as suggested

Comment: L451 use past tense!

Answer: corrected as suggested

Comment: L453 Describe this term

Answer: An explanation has been added.

Comment: Omit this word?

Answer: deleted as suggested

Comment: Explain the difference between these two models

Answer: An explanation has been added in Introduction.

Comment: L465 Omit this sentence

Answer: deleted as suggested

Comment: L467 delete

Answer: deleted as suggested

Comment: L471 Rephrase and explain what this means

Answer: corrected as suggested

Comment: L480 add space

Answer: corrected as suggested

Comment: L481 This figure should be displayed earlier in the paper, along with an explanation as to whether the models also differ in size as well as in the movements of chemicals and organisms

Answer: We have moved figure 10 to the introduction.

Comment: L484 spelling!

Answer: corrected as suggested

Comment: L487 delete

Answer: deleted as suggested

Comment: L494 spelling!

Answer: corrected as suggested

Comment: L495 Replace full stop with a comma

Answer: corrected as suggested

Comment: L504 delete

Answer: deleted as suggested

Comment: L509, Table 2 Suggest delete this Table. The information is expressed more effectively in the Text

Answer: We have removed the table as suggested.

Comment: L515 ?

Answer: corrected as suggested

Comment: L518 use past tense

Answer: corrected as suggested

Comment: L522 rephrase, e.g. insert 'is limited within a'

Answer: corrected as suggested

Comment: L522 delete

Answer: deleted as suggested

Comment: L528 insert of

Answer: corrected as suggested

Comment: L528 replace with 'would accelerate'

Answer: corrected as suggested

Comment: L529 replace with 'causing'

Answer: corrected as suggested

Comment: L530 At an earlier stage you need to decide whether your experiments are meso or macro and give reasons for your choice

Answer: We do not really understand the comment. It was explained in the discussion that some experiments cannot be carried out on the scale of natural experience. Besides, in the case of a flood, it is not possible to prepare a mesocosm - such an experience requires contact with the environment, limited but nevertheless, and in the case of a flood, it is not possible to flood an area without separating it. In the institute we do not have the terrain to simulate such conditions outdoors, in fresh air. For this reason, an experiment was performed under laboratory conditions. However, it seems to me that such information is unnecessary in the paper. In the discussion we refer to model tests. Both mesocosm and microcosm are model experiments. We are comparing their results with the results of experiments under natural conditions. because that is what the work was about - comparing a model experiment with a natural environment.

Comment: L530 U

Answer: corrected as suggested

Comment: L532 e

Answer: corrected as suggested

Comment: L535 omit

Answer: deleted as suggested

Comment: L541 omit 's'

Answer: deleted as suggested

Comment: L546 omit

Answer: deleted as suggested

Comment: L556 insert 'and these'

Answer: corrected as suggested

Comment: L584 omit

Answer: deleted as suggested

Comment: L587 insert 'and flooding'

Answer: corrected as suggested

Comment: L588 omit 's'

Answer: deleted as suggested

Round 2

Reviewer 2 Report

Thank you for comprehensively addressing the points I raised. However, I am disappointed that you have not attempted to make the results easier to understand. Also I dispute your inference that it amounts to plagiarism if you provide a brief description in your own words of a process that you have used as long as it includes a full reference to the originator of the process.